# CP2Image: Generating high-quality single-cell images using CellProfiler representations

**Yanni Ji**                                            2301589j@student.gla.ac.edu
[1] *School of Computing Science, University of Glasgow, Glasgow, Scotland, UK.*

**Marie F.A. Cutiongco**                               Marie.Cutiongco@ntu.edu.sg
[2] *Nanyang Technological University.*

**Bjørn Sand Jensen**                                             bjje@dtu.dk
[1] *School of Computing Science, University of Glasgow, Glasgow, Scotland, UK.*
[3] *Technically University of Denmark, DTU Compute, Kgs. Lyngby, Denmark*

**Ke Yuan**                                             Ke.Yuan@glasgow.ac.uk
[1] *School of Computing Science, University of Glasgow, Glasgow, Scotland, UK.*
[4] *School of Cancer Sciences, University of Glasgow, Glasgow, Scotland, UK.*
[5] *Cancer Research UK Beatson Institute, Glasgow, Scotland, UK.*

**Editors:** Accepted for publication at MIDL 2023

## Abstract

Single-cell high-throughput microscopy images contain key biological information underlying normal and pathological cellular processes. Image-based analysis and profiling are powerful and promising for extracting this information but are made difficult due to substantial complexity and heterogeneity in cellular phenotype. Hand-crafted methods and machine learning models are popular ways to extract cell image information. Representations extracted via machine learning models, which often exhibit good reconstruction performance, lack biological interpretability. Hand-crafted representations, on the contrary, have clear biological meanings and thus are interpretable. Whether these hand-crafted representations can also generate realistic images is not clear. In this paper, we propose a CellProfiler to image (CP2Image) model that can directly generate realistic cell images from CellProfiler representations. We also demonstrate most biological information encoded in the CellProfiler representations is well-preserved in the generating process. This is the first time hand-crafted representations be shown to have the generative ability and provide researchers with an intuitive way for their further analysis.

**Keywords:** Image-based Profiling, Machine Learning, Biological Interpretability.

## 1. Introduction

With the advance in high-throughput microscopy, researchers can efficiently acquire a large number of cell images under a variety of conditions (Wollman and Stuurman, 2007). These images contain rich biological information on cell lineage, biomolecular pathway activation, and morphological cell characteristics, encouraging discoveries focused on visual manifestations of cellular state (Pegoraro and Misteli, 2017).

Several approaches exist to either implicitly or explicitly capture or characterize the morphology of cells (Li et al., 2015; Libbrecht and Noble, 2015; Tatonetti et al., 2012; Wang et al., 2014). Traditionally, cell morphology has been quantified and investigated through the use of hand-crafted features such as CellProfiler (CP) (Carpenter et al., 2006)

or EBImage; however, machine learning approaches have recently shown that it is possible to learn useful representations of cell morphology (Goldsborough et al., 2017) for image generation, phenotype characterization, and downstream prediction tasks (Lafarge et al., 2019). Learned representation often results in impressive performance when generating images of cells (Goldsborough et al., 2017; Lafarge et al., 2019) or discriminating, for example, the mechanism of action (MoA); however, the learned representations (i.e., internal/latent states in a neural network model) are often difficult to understand from a biological perspective as they lack clear linkage with the known biological phenomenon. The use of machine learning for learning representations of cells is therefore partly hindered by the lack of interpretability when it comes to the model's internal, quantitative representation of the cell morphology. On the other hand, many of the machine learning models are so-called generative models with the advantage that they allow scientists an intuitive visual understanding of the representation, thus improving the interpretability of the learned model for non-experts.

Prior work has shown that CP representations contain discriminative information about drug effects and functions, such as the MoA. Hence if we know or make changes to CP representations associated with a specific drug, it would be valuable to visualize the expected effects on cells, thereby facilitating a better understanding of the counterfactuals, out-of-distribution samples, and the overall causal structure between drugs and effects on cells. Furthermore, in rare cases, researchers only have access to CP representations without the corresponding images, thus leaving no opportunity for visual comparison across experiments.

In this work, we propose a new model (CP2Image) that uses CP representations to generate photorealistic images of cells. CP2Image consists of a convolutional neural network-based generator, which takes CP representations as input, and outputs cell images. The model leverages a discriminator network, similar to the VAEplus model (Lafarge et al., 2019), to enhance the image quality. We show that our model generates high-quality cell images from the CP representations. Measured by the Fréchet inception distance (FID) score, one of the most commonly used metrics for measuring synthetic image quality in deep generative model literature (Dhariwal and Nichol, 2021), our model outperforms the state-of-the-art Variational Autoencoder (VAE) (Kingma and Welling, 2013). We also demonstrate how changing the values of input CP representations, such as nuclei-related dimensions, results in corresponding effects in the generated images. We believe this is the first successful attempt at generating photo-realistic single-cell images solely from CP representations.

## 2. Background

The deep convolutional neural network is widely used in extracting single-cell representations and generating images. To extract cell representations and evaluate the representations in downstream analysis such as MoA classification, initial work focuses on the fully supervised methods (Kraus et al., 2016; Godinez et al., 2017) and weakly supervised method (He et al., 2016; Caicedo et al., 2018), but there is a limitation as annotation of images is expensive and time-consuming (Perakis et al., 2021). This encourages researchers to explore other methods. In recent papers, one unsupervised method is introduced that relies on VAE with a discriminator to improve generating performances (Lafarge et al., 2019). Another unsupervised (Janssens et al., 2020) is proposed based on clustering cell images without the need

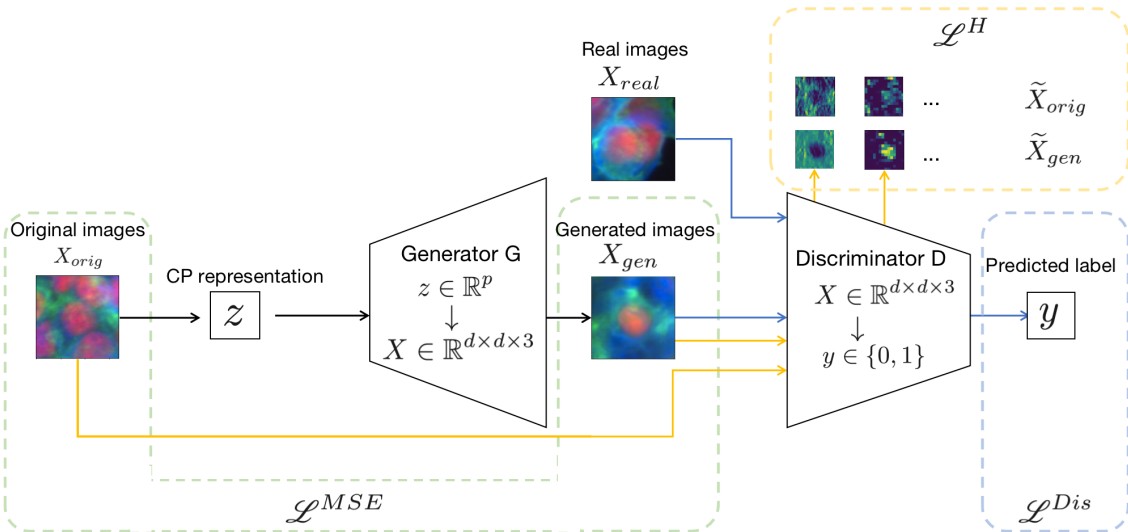

Figure 1: CP2Image model consists of two parts: a generator that generates images from CP representations, and a discriminator to enhance generation performance. The yellow line represents original and generated images feeding into the discriminator for hidden representations MSE loss. The blue line shows the real and generated images feeding into the discriminator for discriminator loss. The model architecture is adapted from VAEplus (Lafarge et al., 2019).

of segmentation of cell images. Contrastive learning is also used to extract representations of cells (Perakis et al., 2021) based on a contrastive loss, which shows competitive performance in MoA classification tasks. In addition, a method incorporating weak labels and a self-supervised knowledge distillation method (Cross-Zamirski et al., 2022) is proposed and achieves state-of-the-art performance in MoA classification. Though the methods using a neural network to extract representations are well-explored, currently, few published studies are exploring exploiting CP representation information for image generation.

## 3. The CP2Image model

To investigate whether interpretable CP representations can be used to generate cell images, we propose a convolutional neural network model, generator $G$, shown in Figure 1. The input is a CP representation $z \in \mathbb{R}^p$, where $p$ is the dimension of a CP feature vector. The output is the generated image $X_{gen} \in \mathbb{R}^{d \times d \times 3}$, where $d$ is the image size. The model consists of four blocks, and each block contains a convolutional layer, a leaky ReLu layer, and an upsampling layer. Batch normalisation layers are used throughout the model. To enhance the generative performance, we add a GAN-style discriminator $D$ as the second component. The discriminator consists of five blocks, and each block contains a convolutional layer, a leaky layer, and a pooling layer. Batch normalisation layers are used throughout the network.

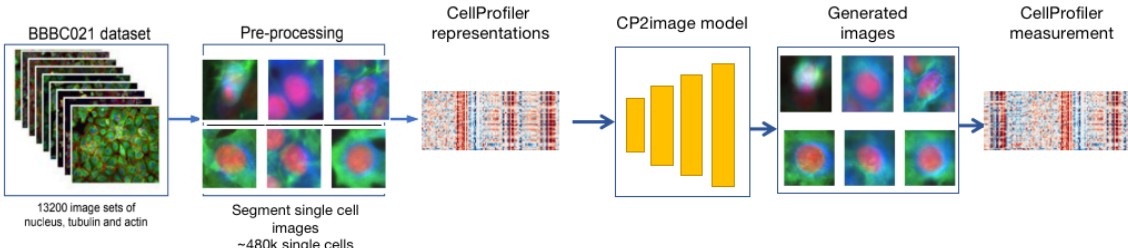

Figure 2: Overview of the experiment: segmentation of single cell images from data pre-processing, extraction of CP representations, images generation via CP2Image model, evaluation of generative performance and downstream analysis.

We first generated images $X_{gen}$ from CP representations via the generator $G$ and calculated the mean squared error (MSE) loss of generated image $X_{gen}$ and corresponding original image $X_{orig} \in \mathbb{R}^{d \times d \times 3}$ as usual loss function, which is denoted as $\mathscr{L}^{MSE}$. To enhance the loss function, we fed the original images and generated images to the discriminator $D$ For each layer $i$, we extracted the hidden representation for that layer and calculated the MSE loss between the original and generated images at that layer, which is denoted as $\mathscr{L}_i^H$. We added up the layers, defined as $\mathscr{L}^H = \sum_i \gamma_i \mathscr{L}_i^H$. This loss encourages the generated images to be similar to the original images at all stages of the process.

Then we fed another real image batch $X_{real} \in \mathbb{R}^{d \times d \times 3}$ into the discriminator $D$, where $X_{real}$ is independent from $X_{orig}$. For real images and generated images, we calculated the cross entropy loss which is used to train the discriminator to classify the real and generated images.

$$\mathscr{L}^{Dis} = -\log D(X_{real}) - \log(1 - D(X_{gen}))$$

We alternate the training between the generator and discriminator during every iteration. When training the generator, we optimize the objective $\mathscr{L}^{MSE} + \mathscr{L}^H$. When training the discriminator, we optimize the objective $\mathscr{L}^{Dis}$. To stabilize the discriminator training, we added spectral normalization (Miyato et al., 2018) in the discriminator network.

## 4. Experiment and results

### 4.1. Experiment

We use the benchmark BBBC021 dataset, which has been used for capturing cell responses to drugs in previous research (Lafarge et al., 2019; Ljosa et al., 2013; Singh et al., 2014; Pawlowski et al., 2016; Ando et al., 2017; Janssens et al., 2020; Caicedo et al., 2018; Qian et al., 2020; Perakis et al., 2021). The dataset is comprised of images of cells that were fluorescently stained against markers for DNA, beta-tubulin and F-actin. A subset of the captured images is labelled as the distinct MoA to represent phenotypic cell effects under compound and concentration. (Smith and Maani, 2001; Ljosa et al., 2013).

Figure 2 shows the overview of the experiment. During the experiment, CP representations are obtained by feeding original BBBC021 cell images into CellProfiler software

(Singh et al., 2014). Every dimension in the CP representations describes an aspect of biological phenotypic information called a feature. The location of the nucleus centre in CP representations is used as the centre of single-cell images during segmentation. To be comparable with other work (Janssens et al., 2020; Lafarge et al., 2019; Perakis et al., 2021), we only keep the MoA annotated subset of 480k single-cell patches and their corresponding CP representations. Then we feed the single-cell images to the CP2Image model for model training. Using a single Titan RTX GPU, it took 11 hours to train a model with 1.32 million parameters on 390k images. After training, it takes an average of 0.02 seconds to generate a single-cell image. Then we evaluate the generating performance by comparing them with VAE, VAEplus and LSGAN models (Goldsborough et al., 2017). To investigate which feature has been retained well in the generating process, we measure CP on generated images and original images, then calculate the pairwise correlation between these two measurements for different features.

## 4.2. Results

To evaluate the generating performance of the CP2Image model and explore its practical usage, we evaluate from three perspectives: compare the generated images with VAEplus, explore the well-preserved information in the generating process and generate conditional cell phenotype.

### 4.2.1. CP representations generate realistic cell images via CP2Image model

To explore the generative performance of the CP2Image model, we generate images via CP2Image and compare them with images generated from VAE and VAEplus. Figure 3 shows four generated images from different models. To quantify the generating performance, we evaluate the images by FID score, with smaller values indicating greater similarity between generated images and real images. Since the Generative Adversarial Networks (GAN) is a prevalent method for image synthesis(Bermano et al., 2022), we also include the LSGAN from (Goldsborough et al., 2017) and compare the resulting FID scores. Table 1 illustrates that CP2Image model can generate less blurry cell images than VAE model, thus CP2Image has superior generative ability compared to the basic VAE. Even though CP2Image has a higher FID score than VAEplus and LSGAN, it should be noted that CP2Image only uses CP representation to produce images, whereas VAEplus and LSGAN generate images using original images and random vectors respectively. The wide availability, interpretability, and predictability of CP representations enable the CP2Image model to help researchers explore and understand the effect of individual CP features/attributes on phenotype, for example, by perturbing an existing CP representation and visualizing the result. This is an immense advantage of the CP2Image model compared with VAE, VAEplus and LSGAN. Figure 3 also indicates that generated images via the CP2Image model have similar nuclei compared to the original images; however, they could not reconstruct the shape of whole cells very well, where the boundaries are sometimes blurry or of irregular shape.

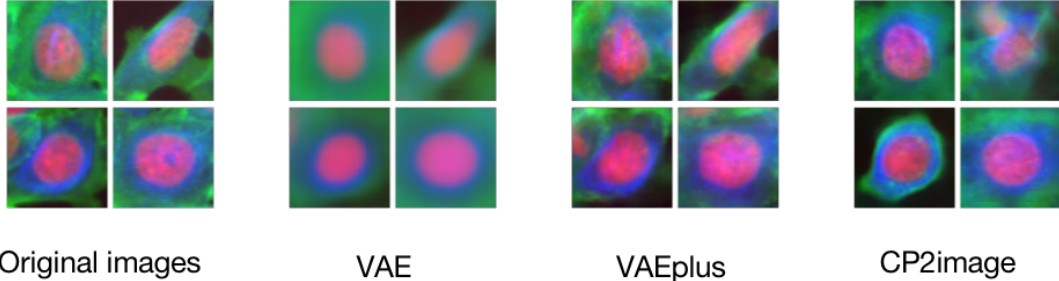

Figure 3: Comparison of original single-cell images and their corresponding images generated by VAE, VAEplus, and the proposed CP2Image model.

Table 1: The FID score is computed by comparing real images and images generated by different models (lower FID is better). The average and standard deviation from three identical trials with random initialization are shown.

|  | VAE | VAEplus | LSGAN | CP2image |
|---|---|---|---|---|
| FID | 351.96±5.3 | 27.98±3.8 | 24.79±2.0 | 68.42±1.9 |

#### 4.2.2. Correlating features from generated images with original images

To further explore if the generated images maintain biological phenotypic information and which feature column in CP representations has been retained in the generating process, we calculate the pairwise correlation of CP measurement between the original and generated images. Figure 4(a) shows the correlation coefficient value, and we can see that more than 30% of feature dimensions have a correlation coefficient larger than 0.5. These highly correlated features demonstrate a large amount of morphological information has been well-preserved during generating process. The 20 features with the largest value are about the shape and intensity of the nucleus, while those features with the smallest values are almost features which describe the distribution of intensity over the single-cell image. Figure 4(b) shows the six dimensions with the highest correlation coefficient values. Figure 4(c) demonstrates when we manipulate a single well-preserved interpretable dimension, the generated images show the corresponding variation. For example, the nuclear area feature measures the number of pixels in nuclei. When we increase its value and fix all other feature dimensions, from left to right, the size of the nucleus gradually increases. The nuclear orientation feature measures the angle between the horizontal axis and the major axis of the nuclei ellipse. Increasing its value allows us to observe a clockwise orientation variation in the generated images.

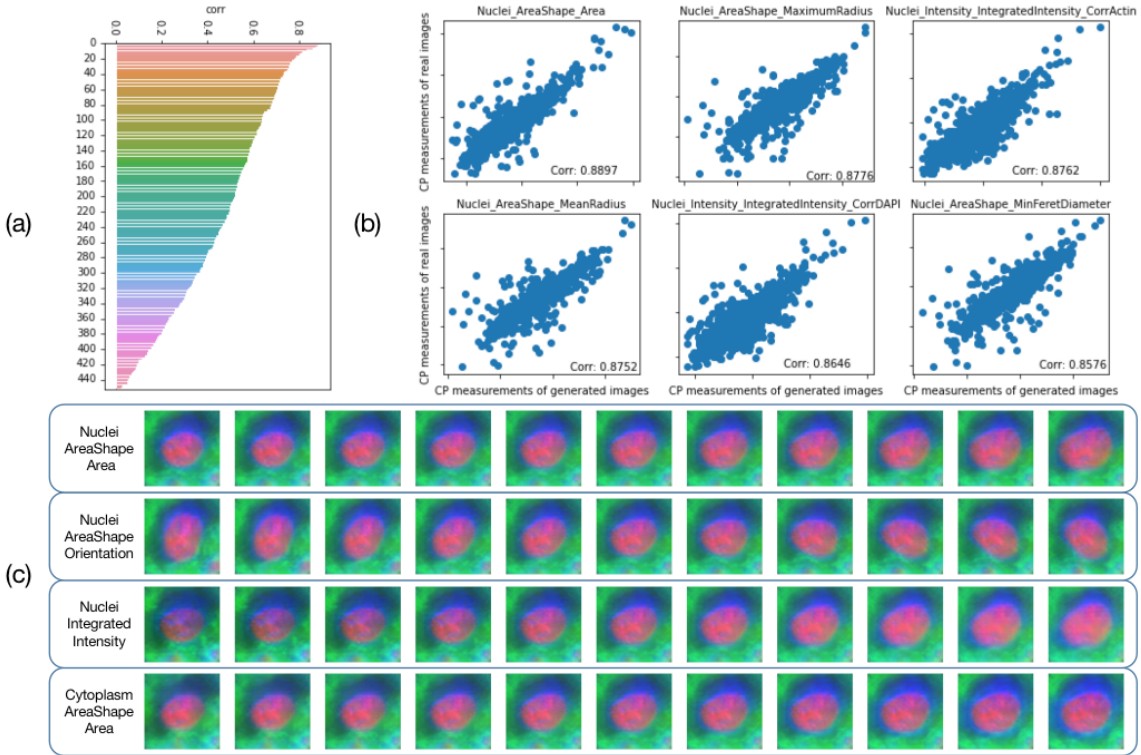

Figure 4: (a) Correlation between the original CP representations of single-cell patches and the CP representation of the CP2Image generated images. (b) Scatter plot of six dimensions of CP representation of single-cell patches and CP measurement with the highest correlation coefficient. (c) Manipulation of a single dimension in the CP representation; increase the value of the nucleus area feature (first row), the nucleus orientation feature (second row), the nucleus intensity feature (third row) and cytoplasm area feature (last row), respectively, and generate images from left to right. We note the clear change of nucleus size in the first row going from left to right, a change of nucleus orientation in the second row, a change of nucleus intensity in the third row and a change of the cytoplasm area in the last row, as expected.

### 4.2.3. Generate conditional phenotype from interpolated CP representations

Here we assess if the model can generate cellular phenotypes that are associated with certain conditions e.g MoA and compound. Conventionally, this would require training conditional generative models. We observe clear discriminative signals in CP representations Figure 5 (b), suggesting CP representations could be treated as embeddings of labels. Therefore, CP2Image could generate specific phenotypes without any conditional fine-tuning. We test

this hypothesis by assessing linear interpolations in CP representations and their corresponding generated phenotypes in both MoA and compound concentrations.

For MoA, the start of the interpolation is set to the negative control, DMSO, which means no compound is used. The end of the interpolation trajectory is set to be the target cell. For each trajectory, DMSO and target cells are chosen as the real images that are closest to the average phenotype under the respective MoA. Every step in the trajectories replaces the values of the CP representations of the DMSO cell with that of the target cell by 50 dimensions. These 50 dimensions are selected with the highest absolute difference between the DMSO cell and the target cell. In Figure 5 (a), we report gradual phenotypic changes from DMSO to the target cell under compounds with three different MoAs in CP2Image-generated trajectories. Particularly, actin disruptors tend to break up the actin cytoskeleton, and in Figure 5 (a), intermediate images show actin (green) gradually disappearing to sporadic dots. We also quantify the similarity of CP representations of interpolated and target cells with cosine similarity, which is shown under each intermediate image. In Figure 5 (a), the cosine similarity achieves 0.95 after only five interpolation steps, and the corresponding generated images are similar to the target cells.

We then show that the generated images from interpolated CP representations could reveal the effect of various compound concentrations on cellular phenotypes. In an experiment similar to (Lamiable et al., 2022), we demonstrate that CP2Image could estimate hard-to-obtain phenotypes. In compound latrunculin B Figure 5 (b), we observe clear clusters of cells treated with different concentrations in a PCA projection of the corresponding CP representations. The proximity of the clusters presents a trajectory from DMSO to the highest concentration $3\mu m$, and a clear gap between the two compounds. We then draw a red line between the mean of DMSO and latrunculin B $3\mu m$, interpolate on the line, and generate images from interpolated CP representations. In Figure 5 (c), a gradual phenotypic evolution matched with the increase of the concentration of the compound.

## 5. Conclusion

In this work, we present CP2Image, a deep generative model to generate high-quality, single-cell images directly from CP representations. We show most biological information is well-preserved in the generated images. Our results demonstrate that hand-crafted representations have the generative capability. Leveraging CP representations being naturally discriminative and interpretable, CP2Image can generate conditional phenotypes without separate training with extra labels, allowing the model to estimate rare and hard-to-obtain phenotypes. This property could be crucial in discovering and characterising novel cellular phenotypes.

Our work can intuitively visualise the properties encoded CP representations. We envision the applications of the CP2Image such as simplifying biological phenomena from datasets with high variability (e.g. inhibitor screening) to help generate new hypotheses or lines of research; ease of visualising repositories with large datasets by providing a *model cell* from CP representations, which can help democratize access and use of imaging datasets (i.e. as an alternative to looking at numbers on a screen); visualises correlation between CP measurements (numerically) and cell morphology (visually), which is useful for inferring

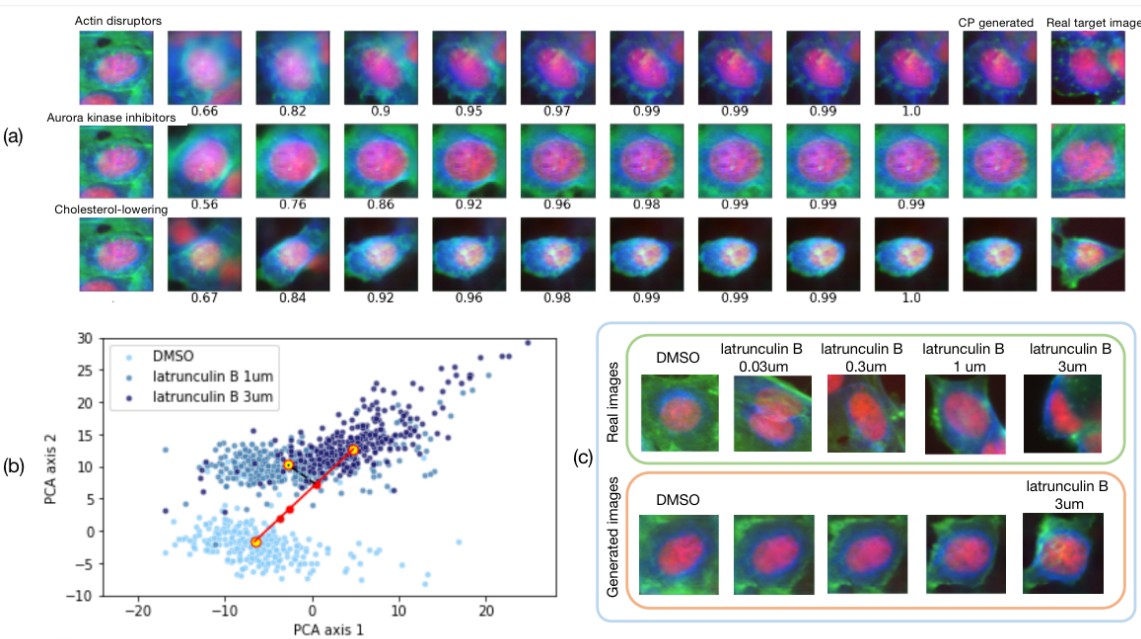

Figure 5: (a) Interpolation of CP representation from a DMSO cell (left) to a target cell (right) from 3 different MoAs, respectively. Every row represents the transitions under the MoA. Every column except the first and last shows the generated images via the CP2Image model. (b) PCA using CP representations of the DMSO, latrunculin B $1\mu m$ and latrunculin B $3\mu m$. (c) Real images were randomly selected from DMSO, four concentrations of latrunculin B (first row), and generated images from interpolated CP representations in the red line between DMSO and latrunculin B $3\mu m$ (second row).

causal relationships between cell treatment/genotype/disease type and morphology. These applications will ultimately help diagnostics and drug screening.

In the future, we plan to test more diverse cell types. Second, we plan to test different types of bio-images where CP representations are applicable. Third, we plan to investigate alternative and emerging generative models such as diffusion models.

## Acknowledgments

We thank all participants and researchers of Broad Institute for providing the BBBC021 dataset.

Yanni Ji acknowledges support from China Scholarship Council from the Ministry of Education of P.R. China. BSJ and KY acknowledge support from the Engineering and Physical Sciences Research Council (EPSRC, EP/R018634/1). KY also acknowledges support from Biotechnology and Biological Sciences Research Council (BBSRC, BB/V016067/1), European Union's Horizon 2020 research and innovation programme project PANCAIM (101016851) and the Wellcome Trust (220977/Z/20/Z).

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
