# OpenReview forum: "CP2Image: Generating high-quality single-cell images using CellProfiler representations"
_MIDL.io/2023/Conference — MIDL 2023 Oral_

### Official Review · Reviewer_TgHW · 2023-02-02

**Confidence:** 3
**Preliminary Rating:** 4

**Summary:**

In this paper an approach to generate realistic cell images out of CellProfiler representations is presented. It follows the idea of Variational Autoencoders, adding a GAN-like discriminator. Evaluation was done using the public BBBC021 data set. According to the results most biological information is well-preserved in the generated images.

**Strengths:**

In general, the contribute of the paper is well defined and it is designed and structured in a concise way.
Being able to generate a large number of highly detailed images with a high similarity in biological features could be a great asset for future research on pathological cellular processes in the context of data hungry deep learning approaches.
In this work well established architectures are combined in an interesting way with hand-crafted experiments allowing to include expert domain knowledge directly into the learning process.


**Weaknesses:**

My major critic with this work is with its actual applicability. I'm still not convinced of the actual use case of this approach. Of course, it is nice to be able to generate a large number of artificial data. To actually use this data to discover novel biological theories, one has to be sure, that all properties are exactly retained, not only some. Even if all properties could be retained, how could you ensure that the generated images represent samples which can be found in real human data?

Due to the generative nature of CP2Image, will the produced images be different even if the same CP representations are fed in? In other words, a large variety of images with the identical CP representation can be generated. If this is the case, is this also reflected in the biological real data?

The state of the art is lacking generative approaches like GANs or Stable Diffusion [1]. At least a discussion on these should be added as GAN-style discriminator is utilized. Of course, a direct comparison e.g. in generative quality to the proposed CP2Image VAE would be highly appreciated.

Please provide a short insight on why FID was used as a similarity metric. There exist several other metrics that could have been used, like the structure similarity index etc. A short comment or even a comparison to other metrics would be appreciated.
VAEplus showed a lower (better) FID score then CP2Image. I cannot fully follow the argument that CP2Image has an advantage since it only uses CP representations. The original images are needed to generate CP representations, so I do not exactly see the advantage here.

Although there are no real time demands in the proposed task, some information on computational performance in inference time, flops or overall number of parameters would be appreciated. These values might seem not so important but are truly relevant when considering algorithms for actual real-world applications. Some information on the hardware used for training/inference would also be interesting.

A discussion on the limitations of the proposed work as well as future perspectives is missing. For example, it is stated that most biological information is well-preserved. Which information is not? Is this important information that is missing?

[1] Rombach et al. “High-Resolution Image Synthesis With Latent Diffusion Models”, in CVPR 2022.


**Deanonymize Review:**

no

**Detailed Comments:**

- Background section:
  - Incomplete sentence. “Another unsupervised (Janssens et al., 2020) is proposed based on clustering cell images without need of segmentation of cell images.”
  - “Though the methods using a neural network to extract representations are well-explored, currently, few published studies are **exploring exploiting** CP representation information for image generation.”
- Table 1: please indicate which values for FID are better, high or low.
- “In Figure 5 (c), a gradual phenotypic evolution matched with the increase of the concentration of the compound.” … is shown
- Figure 5: The variation in the images compared to real images is not preserved. In these examples it seems like mainly the background is changed. can you elaborate on that?


**Paper Type:**

validation/application paper

**Questions To Address In The Rebuttal:**

As already stated in the weakness part, I'd like the authors to include a discussion on the biological and/or clinical implications of their work. Furthermore, an improvement of the state-of-the-art section including some recent approaches for image synthesis is needed. Finally, a discussion on the limitations and future directions should be added.

---

### Official Review · Reviewer_1CWQ · 2023-02-04

**Confidence:** 3
**Preliminary Rating:** 5
**Recommendation:** Oral

**Summary:**

The manuscript proposes a CellProfiler to Image (CP2Image) model that can directly generate realistic cell images from CellProfiler representations. They also demonstrate that most biological information encoded in the Cell Profiler (CP) representations is well-preserved in the generating process. They demonstrate how changing the values of input CP representations, such as nuclei-related dimensions, results in corresponding effects in the generated images. They demonstrate that hand-crafted representations have the generative ability and provide the intuition to facilitate further analysis.

**Strengths:**

Strengths –
-	Direct practical application
-	Intuitive method
-	CP2Image can generate conditional phenotypes without separate training with extra labels, allowing the model to estimate rare and hard-to-obtain phenotypes.



**Weaknesses:**

Weaknesses
-	The generated images via the CP2Image model have similar nuclei compared to the original images; however, they could not reconstruct the shape of the whole cells very well, where the boundaries are sometimes blurry or of irregular shape.



**Deanonymize Review:**

no

**Paper Type:**

validation/application paper

**Questions To Address In The Rebuttal:**

Questions to address in the rebuttal

-	What could be the cause of the boundaries being blurry or of irregular shape? How can it be addressed in the future?
-	The visual quality of VAEplus images is better than the one of CP2Image. What are the advantages of CP2Image in comparison with VAEplus?

---

### Official Review · Reviewer_GqvC · 2023-02-04

**Confidence:** 4
**Preliminary Rating:** 5
**Recommendation:** Oral

**Summary:**

Single-cell high-throughput microscopy images can represent various biological information such as hand-crafted features CellProfiler (CP). However, those learnable representations are hard to understand. Therefore, the author proposed CP2Image to generate an intuitive visual understanding of the representation (photorealistic images of cells) from the CP representation.

**Strengths:**

(1) This is the first approach to generate photo-realistic single-cell images with good reconstruction results from CP representations, compared with VAEs.

(2) Explicated image analysis by feature comparison from the original images and generated images. The visualization of the results is clear. The visualization of manipulating a single interpretable dimension shows the functionality of the proposed model between Cps to Images.

(3) The external validation demonstrates that CP2Image generates different phenotypes without any conditional fine-tuning.

**Weaknesses:**

(1) Figure 1 is confusing without any loss function.

(2) It could be helpful if the author provides the best/median/worst cases of baseline models to further understand the generation performance.

(3) The other SOTA GAN-based/diffusion model are encouraged to compare.

**Deanonymize Review:**

no

**Detailed Comments:**

(1) Why X_real is independent from X_orig? Why was the MSE loss used twice between X_orig and X_gen, and between hidden representation from X_gen and X_real?

(2)Were L_mse, L_H, and L_Dis weighted (with hyper-parameters) during the training?

**Paper Type:**

methodological development

**Questions To Address In The Rebuttal:**

Please address the weakness and detail comments proposed above. It would be helpful if the author could provide analytics on generated images, as well as a performance comparison between SOTA GAN-based/diffusion model in the paper.

---

### Meta-Review · Area_Chair_XuGm · 2023-02-22

**Recommendation:** Accept (Oral)
**Confidence:** 4

**Metareview:**

This paper proposes a generative model for going from CellProfiler representations to realistic cell images. Going from hand-crafted features to images is an interesting avenue for research that has been less explored. It has the advantage of being potential more interpretable over standard GAN-like approaches. All reviewers were in agreement that the presented work is innovative and of value for the MIDL community. The only major criticism from the reviewers were lack of a demonstration of a use-case for the proposed method; however, considering the novelty of the approach more complete experiments can be left for a follow-up work.